# BehaveNet: nonlinear embedding and Bayesian neural decoding of behavioral videos

**Eleanor Batty\*, Matthew R Whiteway\*, Shreya Saxena, Dan Biderman, Taiga Abe**
Columbia University
erb2180,m.whiteway,ss5513,db3236,ta2507 @columbia.edu

**Simon Musall**
Cold Spring Harbor
simon.musall@gmail.com

**Winthrop Gillis**
Harvard Medical School
win.gillis@gmail.com

**Jeffrey E Markowitz**
Harvard Medical School
jeffrey_markowitz@hms.harvard.edu

**Anne Churchland**
Cold Spring Harbor
churchland@cshl.edu

**John Cunningham**
Columbia University
jpc2181@columbia.edu

**Sandeep Robert Datta**
Harvard Medical School
srdatta@hms.harvard.edu

**Scott W Linderman**[†]
Stanford University
swl1@stanford.edu

**Liam Paninski**[†]
Columbia University
liam@stat.columbia.edu

## Abstract

A fundamental goal of systems neuroscience is to understand the relationship between neural activity and behavior. Behavior has traditionally been characterized by low-dimensional, task-related variables such as movement speed or response times. More recently, there has been a growing interest in automated analysis of high-dimensional video data collected during experiments. Here we introduce a probabilistic framework for the analysis of behavioral video and neural activity. This framework provides tools for compression, segmentation, generation, and decoding of behavioral videos. Compression is performed using a convolutional autoencoder (CAE), which yields a low-dimensional continuous representation of behavior. We then use an autoregressive hidden Markov model (ARHMM) to segment the CAE representation into discrete "behavioral syllables." The resulting generative model can be used to simulate behavioral video data. Finally, based on this generative model, we develop a novel Bayesian decoding approach that takes in neural activity and outputs probabilistic estimates of the full-resolution behavioral video. We demonstrate this framework on two different experimental paradigms using distinct behavioral and neural recording technologies.

Understanding the complex relationship between neural activity and behavior requires a thorough characterization of behavior across multiple timescales. Behavior has traditionally been characterized by low-dimensional, task-related variables such as reaction times, or the position of a joystick, or the speed of a wheel turn. These require specialized sensors set up by the experimenter, necessitating laborious testing and calibration.

Of course, behavior is in reality potentially very high-dimensional, and there is a growing appreciation that to understand neural activity we need to monitor behavior in a less simplistic (and labor-intensive)

way [1, 2, 3, 4]. There has recently been a growing interest in automated analysis of video data collected during experiments, aimed at extracting richer, higher-dimensional representations of behavior. For example, the last couple years have seen dramatic improvements in markerless tracking of body parts [5, 6, 7]. These tracking methods have opened up a range of exciting new studies, but come with some drawbacks. Tracking methods are supervised, and therefore require user effort to label training images. Furthermore, simply tracking a few body parts may not capture all of the useful information in the video. For example, subtle changes of facial expression or body pose may be difficult to track with a few markers. Moreover, the tracked landmarks are chosen by the experimenter, and as such, important variables may potentially be missed. Another drawback to tracking methods is that occlusion or movement out of frame may cause markers to be dropped; if downstream analyses do not properly handle missing data these frames must be excluded from analysis.

In a separate thread of work, fully-unsupervised linear dimensionality reduction methods such as Singular Value Decomposition (SVD) have been used to analyze behavioral videos [8, 9], but these approaches require a large number of dimensions to represent behavioral videos (typically, >200 dimensions are chosen), potentially hampering downstream analyses. In fact, we have no reason to expect that images of a moving animal can be represented in a low-dimensional linear vector space, as required for SVD to be an effective model.

Once low-dimensional time series corresponding to behavior have been obtained — whether through supervised or unsupervised methods — we would like to model the dependence of neural activity on these behavioral signals, after characterizing this behavior at different timescales. Most previous approaches have focused on directly mapping the extracted signals into neural activity, i.e., fitting "encoding models" that predict neural responses from the observed behavioral signals [8, 9]. However, a number of alternative analysis approaches are possible [10, 11], including unsupervised modeling of the full behavioral video [12], decoding behavior directly from neural signals [13, 14, 15], or jointly modeling both the behavior and neural signals [16, 17].

Here we introduce a probabilistic framework for the unsupervised analysis of behavioral video, with semi-supervised decoding of neural activity. This framework provides tools for compression, segmentation, generation, and decoding of behavioral videos. Compression is performed using CAEs, which yield a low-dimensional, continuous representation of behavior that requires fewer dimensions than linear methods (e.g., SVD) to obtain the same video reconstruction error. We then use an ARHMM to segment the CAE representation into discrete "behavioral syllables." The resulting generative model can be used to simulate behavioral video data. Finally, we exploit this generative model to construct Bayesian decoders which take in neural activity and output probabilistic estimates of the full-resolution behavioral video. We demonstrate the use of this framework using two popular experimental paradigms and neural recording technologies: Neuropixel multielectrode array recordings during spontaneous behavior in head-fixed mice [9, 18], and widefield calcium imaging from task-engaged head-fixed mice [8, 19].

**Related work**. We build upon a rich literature of behavioral analysis. Stephens et al. [20] showed that the posture of the nematode *C. elegans* is captured by a low dimensional subspace of "eigenworms." Studies have been performed on pose and posture estimation in other model organisms with similar results [21, 22, 23]. Our work is inspired by Wiltschko et al. [24] on characterizing mouse behavior. In this work, behavioral videos of freely behaving mice are compressed using PCA, followed by segmentation via ARHMMs. Using these models, the authors identify behavioral syllables in mouse behavior such as rearing and grooming. Extending this work, Johnson et al. [12] combined compression and time series modeling in a structured variational autoencoder. Related models have been developed for other model organisms, including *C. elegans* [25, 26] and larval zebrafish [27]. Recently, Markowitz et al. [28] has used these methods to identify specific neural representations of behavioral syllables.

Parallel advances have been made in the analysis of neural time series. Sequential variational autoencoders [29, 30] and recurrent state space models [31, 32, 33] capture low dimensional structure in neural activity and relate it to sensory inputs and motor outputs. These build on a long line of work decoding movement from neural activity which we do not have room to adequately review here [13, 14, 15]. Most of these approaches have a low-dimensional output that consists of either the electromyography (EMG) of a handful of muscles in the limb, or the kinematics of the end-effector, or both. They do not capture the high-dimensional facial movements or bimanual arm poses that are the focus of this study.

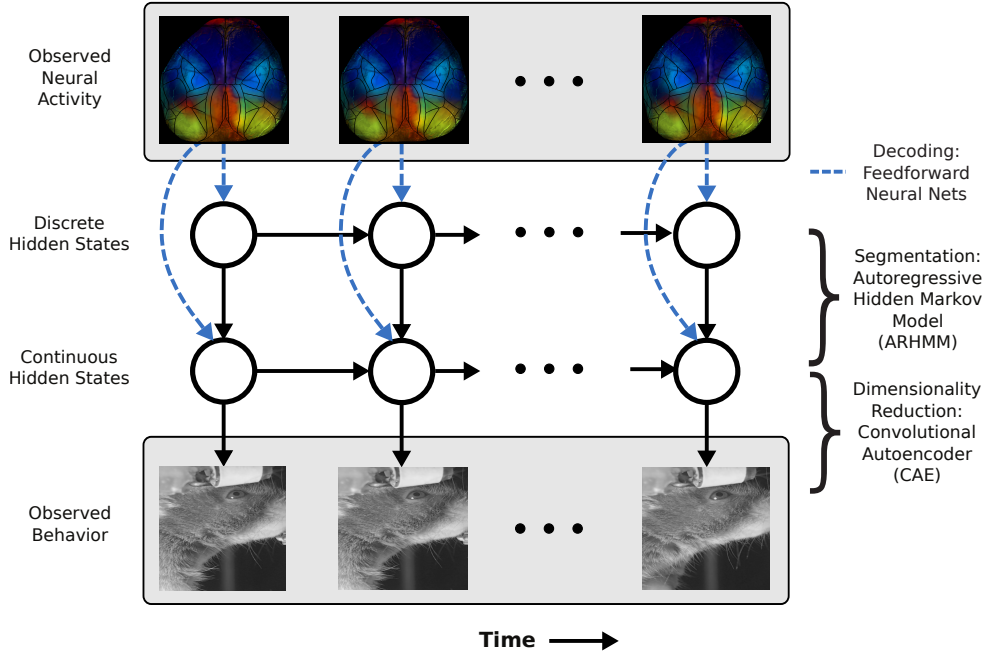

**Figure 1:** Graphical model showing the architecture we use for the neural decoding of continuous and discrete states estimated directly from the behavior. Example neural and behavior data shown for the WFCI dataset, as detailed in the text.

Finally, outside of the motor decoding literature, a couple threads of work on image and speech decoding from neural signals are particularly relevant: Parthasarathy et al. [34] developed approximate neural network Bayesian decoding methods to decode high-dimensional natural images directly from populations of retinal ganglion cells, while Akbari et al. [35] and Anumanchipalli et al. [36] used structured neural network approaches to decode high-resolution speech signals from neural activity.

## Methods

We begin by describing the datasets used in this work (data splits are described in Appendix A). Then we describe the methods used for compression, segmentation, and decoding.

*Widefield Calcium Imaging (WFCI) dataset* [8, 19]. A head-fixed mouse performed a visual decision-making task while neural activity across dorsal cortex was optically recorded using widefield calcium imaging. We used the LocaNMF decomposition approach to extract signals from the calcium imaging video [37]. Behavioral data was recorded using two cameras (one side view and one bottom view; Fig. 2B, *left*); grayscale video frames were downsampled to 128x128 pixels. Data consists of 1126 trials across two sessions, with 189 frames per trial (30 Hz framerate). Neural activity was acquired at the same frame rate.

*Neuropixels (NP) dataset* [9, 18]. A head-fixed mouse behaved freely (including spontaneous manipulation of a wheel with its forelimbs) while neural activity across multiple brain structures was electrically recorded using eight Neuropixels probes [38]. Behavioral data was recorded using a single camera (Fig. 2B, *center*); grayscale video frames were downsampled to 112x192 pixels. Data consists of 96k frames (40 Hz framerate), and "trials" were arbitrarily defined as blocks of 1000 frames. Neural activity was binned at the video frame rate.

*Neuropixels-zoom (NP-zoom) dataset.* We cropped the behavioral videos in the NP dataset in order to analyze the fine-grained facial movements of the mouse (Fig. 2B, *right*); grayscale video frames were downsampled to 128x128 pixels after cropping, and the bottom corners were masked to occlude forelimb movements near the face.

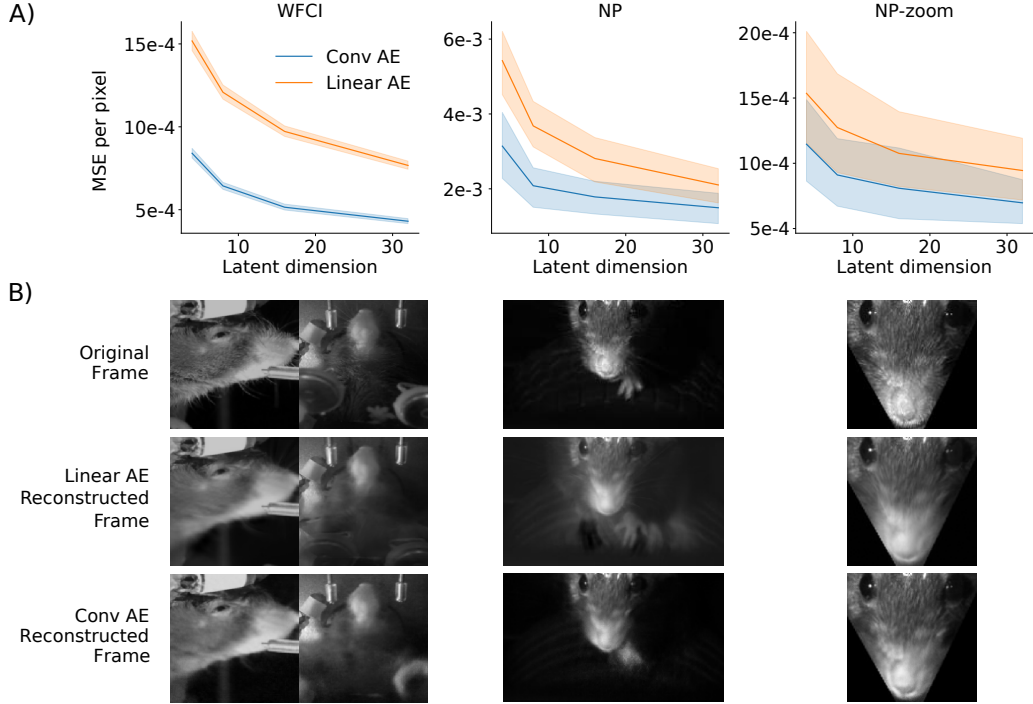

**Figure 2:** The CAE obtains good reconstructions at high compression rates. A) Reconstruction MSE on held-out test data as a function of latent dimension. The nonlinear CAE consistently outperforms the linear autoencoder. Plotted values are means over test trials, and errorbars represent 95% bootstrapped confidence intervals. B) Reconstruction quality is good even when the latent dimensionality is three orders of magnitude smaller than the original number of pixels per frame. Top row shows example original frames from held-out test data in each dataset using 8 CAE dimensions; middle and bottom rows show corresponding linear autoencoder and CAE output frames, respectively. In the WFCI bottom view we have enhanced the contrast and clipped high pixel values in all figures for better visibility. Also see Supplementary Videos C.1 for full reconstruction videos.

**Nonlinear dimensionality reduction of behavioral videos**. We compress the behavioral videos with a convolutional autoencoder (CAE), yielding a low-dimensional continuous representation of behavior that is useful for downstream analyses. The CAE architecture is fixed for all datasets, except for the number of latents (Fig. 2; see Appendix A for architecture details). We train the autoencoders by minimizing the mean squared error (MSE) between original and reconstructed frames using the Adam optimizer [39] with a learning rate of $10^{-4}$. Models are trained for a minimum of 500 epochs and a maximum of 1000 epochs. Training terminates when MSE on held-out validation data, averaged over the previous 10 epochs, begins to increase. As a baseline comparison we also fit a linear SVD model[1].

**Segmentation of behavior**. The CAE outputs a low-dimensional nonlinear embedding of the behavioral video frames, but does not capture temporal dependencies between frames. Next we train a simple class of nonlinear dynamical systems to approximate dynamics within this embedded space.

Let $x \in \mathbb{R}^{T \times D}$ denote the sequence of continuous latents obtained by embedding the video frames with the CAE. Each latent $x_t$ corresponds to the embedding of the corresponding video frame at timestep $t$; $T$ is the video length and $D$ is the embedding dimension ($D$ is of order 10 in the examples here). Building on previous work [24, 28, 40, 41, 42], we model the sequence of continuous latents as a stochastic process that switches between a small number $K$ of discrete regimes, each characterized by linear-Gaussian dynamics. These discrete regimes are specified by an additional layer of discrete

state variables $z \in \{1, \ldots, K\}^T$, and they too may exhibit temporal dependencies; the discrete state at time $t$ may depend on its preceding value. These modeling assumptions are captured by an autoregressive hidden Markov model (ARHMM), which specifies a joint distribution over continuous and discrete state sequences,

$$p(x, z; \theta) = p(z_1) \, p(x_1) \prod_{t=2}^{T} p(z_t \mid z_{t-1}; \theta) \, p(x_t \mid x_{t-1}, z_t; \theta)$$

$$= \pi_{z_1} \, \mathcal{N}(x_1 \mid \mu_1, \Sigma_1) \prod_{t=2}^{T} P_{z_{t-1}, z_t} \, \mathcal{N}(x_t \mid A_{z_t} x_{t-1} + b_{z_t}, Q_{z_t}), \tag{1}$$

where $\pi \in \Delta_K$ specifies the initial distribution over discrete states, $(\mu_1, \Sigma_1)$ parameterize a Gaussian initial distribution over continuous states, $P \in [0, 1]^{K \times K}$ is a row-stochastic transition matrix, and the parameters $\{A_k, b_k, Q_k\}_{k=1}^{K}$ specify the linear dynamics associated with each of the $K$ discrete states. These parameters are combined in the set $\theta = \{\pi, \mu, \Sigma, P, \{A_k, b_k, Q_k\}_{k=1}^{K}\}$.

We fit the ARHMM with expectation-maximization (EM). As in standard hidden Markov models [43], the posterior expectations in the E-step are obtained via message passing in the chain-structured discrete graphical model. The optimal dynamics parameters are found via weighted least squares regression. We present results with ARHMMs with a single autoregressive lag.

The fitted ARHMM produces a discrete segmentation of the sequence of continuous latents output by the CAE. We estimate the discrete states with the maximum a posteriori (MAP) state sequence $z^* = \arg\max_z p(z \mid x, \theta^*)$, which we obtain via the Viterbi algorithm. The estimated state sequence serves multiple purposes. As we will see, the discrete states may offer useful interpretations of behavior as a sequence of discrete "syllables," patterns of behavior identified by similar temporal dynamics. Moreover, different discrete states may correspond to different patterns of neural activity, and different mappings from neural activity to continuous latent states. We will leverage this feature of the discrete segmentation when developing the Bayesian decoders next.

**Decoding behavior from neural activity**. Our ultimate goal is to develop a clearer understanding of how neural activity maps to observed behavior (and vice versa). Probabilistic models of behavior offer a useful means to that end. Specifically, probabilistic models like the ARHMM offer a set of latent states that summarize behavioral time series, and thus a low-dimensional target for neural decoding. We develop a nonlinear Bayesian decoder that combines neural recordings with the ARHMM prior to yield a posterior distribution over behavioral videos given neural activity.

Ideally, we would learn the likelihood of the observed neural activity $u \in \mathbb{R}^{T \times N}$, where $N$ is the number of neurons, given the underlying discrete states $z$ and continuous states $x$. With a good likelihood model, we could combine it with the ARHMM prior to obtain a posterior distribution $p(x, z \mid u)$ for our Bayesian decoder. Unfortunately, learning a good likelihood model is challenging, so we take an alternative approach in order to sidestep this problem.

Inspired by Burkhart et al. [44], we instead train feedforward neural networks to output conditional distributions $p(z_t \mid u_{t-\Delta:t+\Delta})$ and $p(x_t \mid u_{t-\Delta:t+\Delta})$ over the discrete and continuous states, respectively, given a window of neural activity. These are trained discriminatively, using the latent states inferred from the behavioral data. Details of the architecture, training procedure, and hyperparameter searches are in Appendix A.

We use Bayes' rule to write $p(u_{t-\Delta:t+\Delta} \mid z_t) \propto p(z_t \mid u_{t-\Delta:t+\Delta})/p(z_t)$, where the proportionality constant $p(u_{t-\Delta:t+\Delta})$ is constant with respect to $z_t$. The numerator is given by the feedforward networks and the denominator is the marginal distribution under a Markov chain, which for long sequences is well-approximated by the stationary distribution. We plug in this ratio as a substitute for the likelihood in a hidden Markov model, and then use standard message passing routines to sample and compute expectations of $z$ under the posterior $p(z \mid u)$. Of course, this is the posterior distribution under an approximate model of $p(u \mid z)$; nevertheless, it suffices for combining the ARHMM prior and the neural data in a Bayesian way.

We use the same technique to obtain posterior samples of the continuous states $x$, but here we condition on both the neural data and a sample $z \sim p(z \mid u)$. Here, we need the marginal distribution $p(x_t \mid z)$, which we obtain from a simple Kalman filter with time-varying dynamics parameters determined by $z$. Given the marginal distribution and the conditional distribution output by the neural network, we use the Kalman smoother to compute posterior expectations of the continuous latent states. In

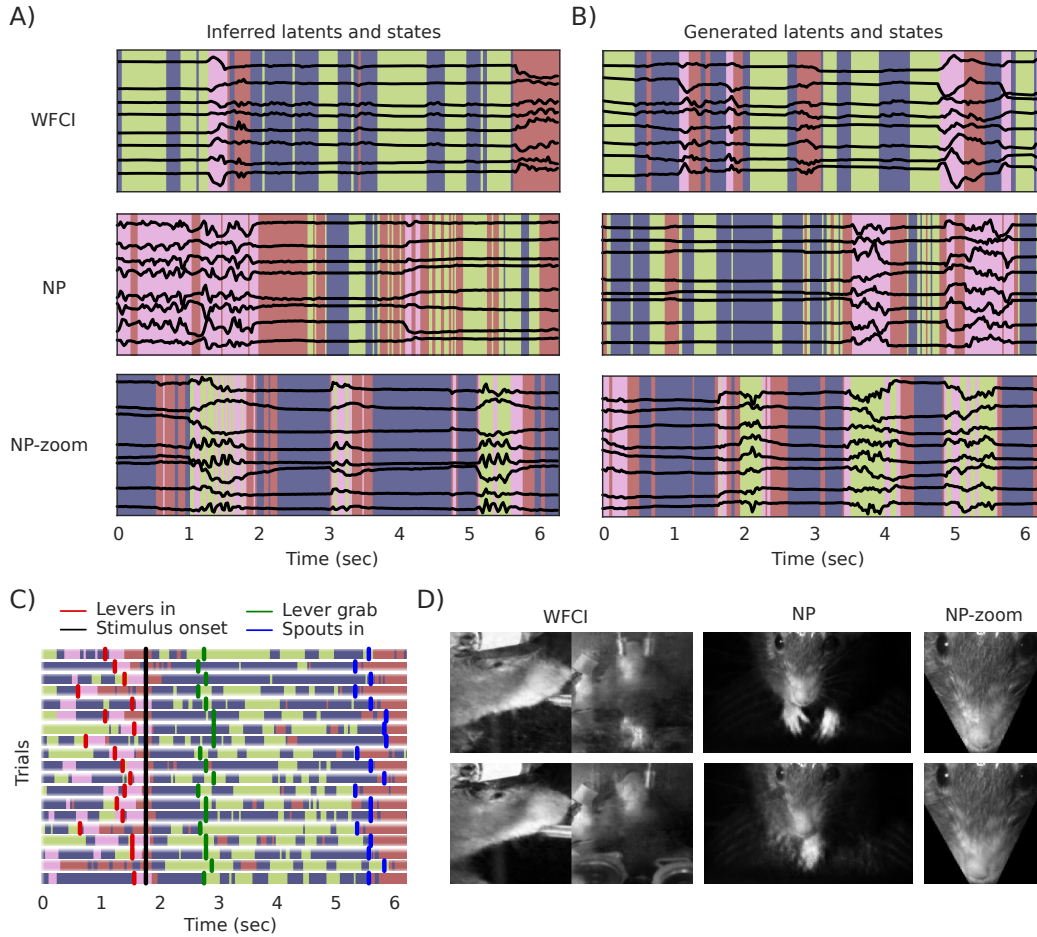

**Figure 3:** Segmenting behavioral traces and sampling new traces and videos from the generative model. A) CAE latents are shown on held out test data over time, with background colors indicating the discrete state (K=4) inferred for that time step using the ARHMM (colors are chosen to maximally differentiate states; colors do not indicate the same states across different datasets). Transitions from rest to movement are easily detectable based on the assigned colors. B) Similar to A) but the latents and states are generated by sampling from the ARHMM; resulting traces are qualitatively similar to real traces, with similarly strong heterogeneity in smoothness in different temporal segments. C) Discrete states are shown for 19 trials of the WFCI dataset, aligned to a right lever grab in the behavioral task. The same states (labeled by the same colors as in A) and B) above) frequently occur at similar points in each trial, indicating trial-locked state structure. Trial specific time points such as the levers moving in and stimulus onset are overlaid. D) Two random frames from videos sampled from the full generative model learned for each dataset; in each case the generated frames resemble real frames. See Supplementary Videos C.4 for full generated videos.

doing so, we obtain a Bayesian estimate of the discrete and continuous states given the neural data. Finally, given sample sequences $x_{1:T}$ we can again map these sequences through the CAE decoder to obtain full videos $y_{1:T}$ sampled from the posterior.

## Results

**Nonlinear dimensionality reduction**. We begin by quantifying the performance of the CAE. The critical result here is that the behavioral videos can be embedded in a low-dimensional space (Fig. 2): an embedding dimension $D < 20$ suffices to capture much of the structure visible in the mouse's behavior (though unsurprisingly very high-resolution details such as the tips of the whiskers are blurred at this level of compression). Even linear autoencoders achieve decent compression, though the nonlinear CAE outperforms the linear model consistently, particularly in frames where large paw

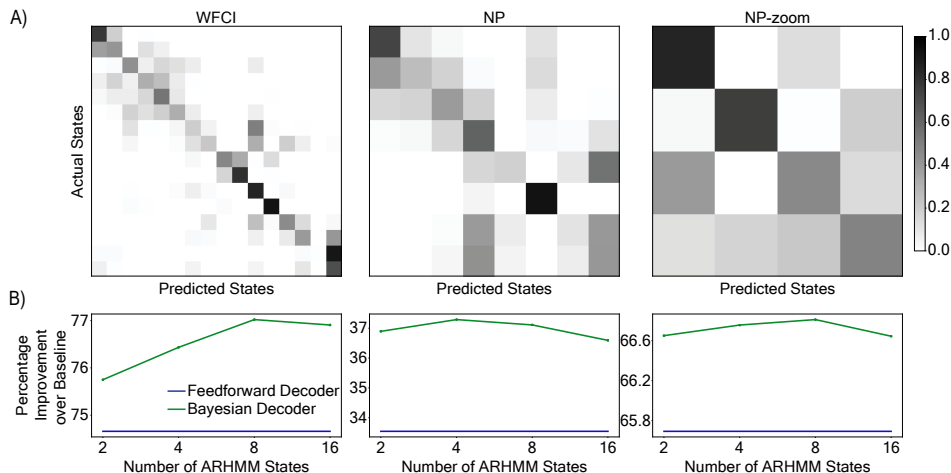

**Figure 4:** Summary of decoding results. A) Confusion matrices of predicted vs actual inferred discrete states for each dataset for the Bayesian decoder. A diagonal matrix corresponds to perfect decoding. States ordered by usage in training data. The Bayesian decoder and feedforward decoder (not shown) perform similarly, both outperforming chance. B) Both the Bayesian decoder and the feedforward decoder outperform baseline for CAE latent predictions from neural activity. The percent improvement over baseline is shown as a function of the number of discrete states used by the ARHMM prior for the Bayesian decoder.

motions occur (see Supplementary Videos C.1 for reconstructions). Importantly, for the WFCI dataset, the CAE operates on both camera views simultaneously, allowing us to combine information from multiple sources into one low-dimensional representation. Throughout the rest of the paper, we will use CAEs with an 8-dimensional latent space for each dataset.

**Segmentation of behavioral video**. We fit ARHMMs to segment the behavior based on the dynamics of the CAE latents. The segmentation corresponds to visible changes in the CAE latents (Fig. 3A). With two ARHMM states, we segment the behavior roughly into moving vs still for all datasets (see Supplementary Videos C.2 with K=2). With an increased number of ARHMM states, we see more nuanced segmentation. We examined the reproducibility of these segmentations across trials in the WFCI dataset; clear trial-locked state structure is visible in Fig. 3C, indicating that these models are capturing reproducible structure in the CAE latents. We also examined the reproducibility of these segmentations across mice (Fig. A2), and find that a similar trial-locked state structure is shared across multiple animals.

**Sampling from the full generative model**. The ARHMM fit to data serves as a generative model of behavioral videos. First, we sample forward from the ARHMM with learned parameters $\theta^*$ and most likely state sequence $z^*$ to obtain continuous state sequences $x_{1:T}$ (Fig. 3B). We then feed the sequence of continuous latents into the CAE decoder to obtain novel synthetic behavioral videos $y_{1:T}$ (see Supplementary Videos C.4). The resulting generative process is clearly not perfect; there are occasional distorted frames, and given enough viewing time it is easy for human observers to distinguish real versus sampled movies. Nevertheless, many generated frames qualitatively resemble real frames (Fig. 3D) and we can see the mouse transitioning between still and moving in a fairly natural way in the sampled videos, indicating that the generative model places significant probability mass near the space of real behavioral videos.

**Bayesian decoding of behavioral states, latents, and full videos**. We use this generative ARHMM model as the basis of a fully Bayesian decoder that operates on neural activity to reconstruct mouse behavior. We first fit separate feedforward neural network decoders to predict discrete states and CAE latents, and then incorporate these decoders into a fully Bayesian decoder (see Methods). We choose the number of states for the ARHMM based on Bayesian CAE mean squared error on validation data (WFCI: 16 states, NP: 8, NP-zoom: 4). We compare decoding performance to baseline predictions which are defined as the most common state (discrete decoder) and the mean value of the CAE latents (continuous decoder) on training data.

Decoder predictions of both continuous CAE latents and discrete ARHMM states are above chance across datasets (Fig. 4). Consistent with previous work [8, 9], we find that the neural signals recorded in these experiments contain rich information about behavior. For discrete states, the confusion

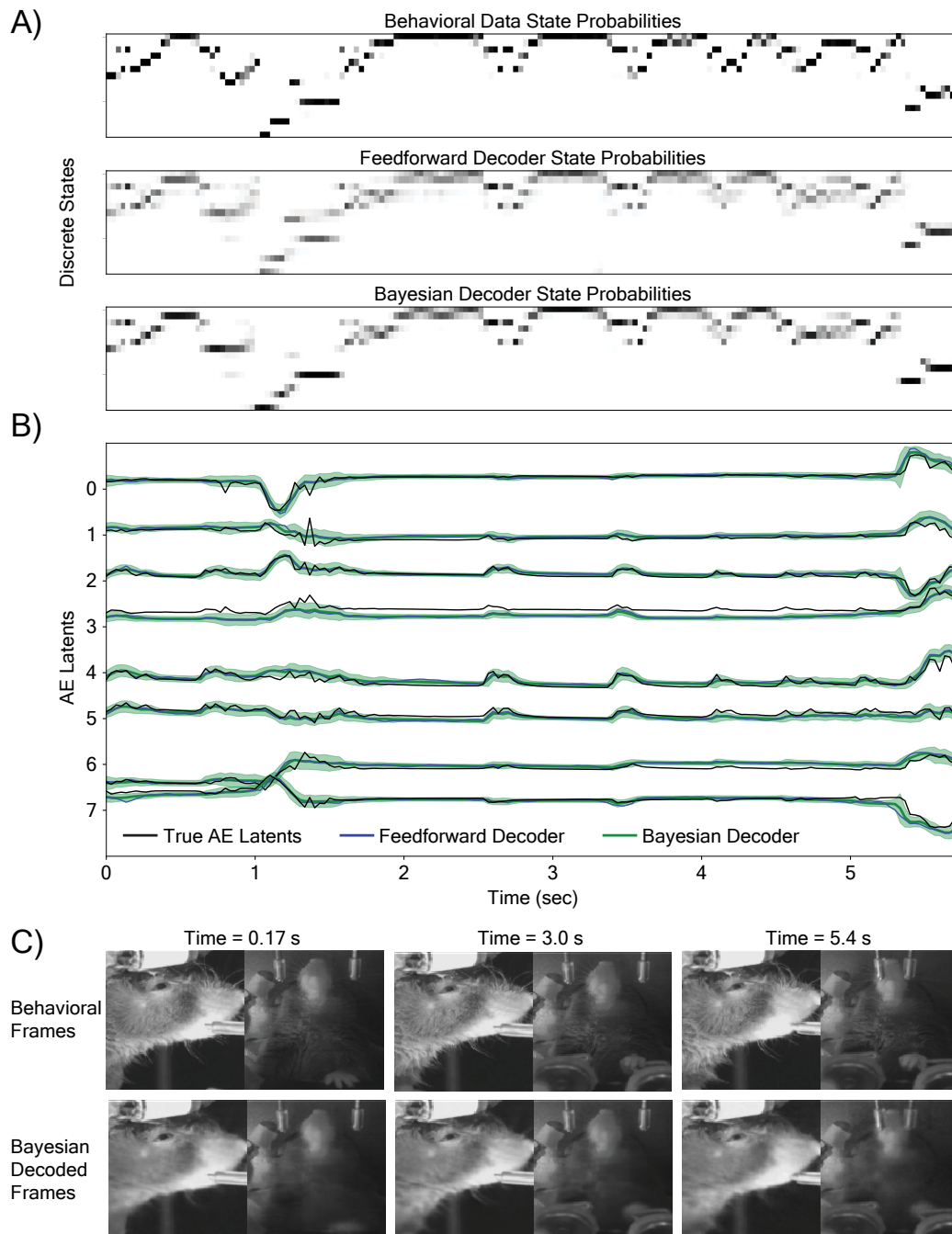

**Figure 5:** Example decoded ARHMM states and CAE latents for a held out WFCI test trial. A) Discrete state probabilities inferred by the ARHMM for the behavioral data (*top row*); predictions from a feedforward model operating on neural data (*middle row*); output of the Bayesian decoding model (*bottom row*). B) CAE latents (*black*) are compared to predicted latents from a feedforward decoder (*blue*) and the Bayesian decoder (*green*). The shaded region indicates ±3 posterior SDs, output by the Bayesian posterior. C) Example frames of the real behavioral video (*top row*) compared to the Bayesian decoded frames (*bottom row*); see Supplementary Videos C.3 for full details.

matrices of actual vs predicted states (Fig. 4A), sorted by usage of the actual states in the training data, show a diagonal structure, reflecting above-chance performance (WFCI: 53% correct vs 19% baseline for 16 states, NP: 53% correct vs 32% baseline for 8 states, NP-zoom: 67% correct vs 37% baseline for 4 states; feedforward decoding accuracy is comparable to Bayesian).

For the continuous latents, the Bayesian decoder improves over baseline MSE for latent trace prediction by 77% for WFCI and 67% in the NP-zoom dataset (Fig. 4B). The Bayesian decoder slightly outperformed feedforward decoders; feedforward decoders in turn outperformed simple linear decoders (results not shown). The Bayesian decoder offers less improvement (37%) in the NP dataset, which may seem surprising since the NP-zoom behavioral video is simply a crop of the NP video. Our interpretation is that the variance in the CAE latents extracted from the NP video are dominated by the location of the mouse's paws in space, and the brain regions recorded from in this experiment carried much more information about the discrete states and the facial pose than absolute paw location. Future work analyzing data from a richer variety of brain regions will further test this hypothesis.

Finally, Fig. 5 shows the feedforward and Bayesian decoder predictions for discrete states and CAE latents for an example test trial from the WFCI dataset. See Supplementary Figs. A5 and A6 for example trials for NP-zoom and NP datasets. The Bayesian decoder also provides valuable information at each time step about the uncertainty of the CAE latents — this information is not directly available from the feedforward decoder. In Supplementary Videos C.3 we show several samples of the full decoded video next to the real video, to provide a more detailed illustration of the posterior variability.

## Discussion

We have introduced a framework for the compression, segmentation, generation, and decoding of behavioral videos. Our approach builds on previous work that used ARHMMs to segment behavioral videos [24, 28]. We extend these methods by incorporating nonlinear autoencoders (providing more accurate and compact representations of the video signal) and introducing a novel Bayesian decoding approach that exploits this ARHMM prior backbone; the resulting generative model and decoder can output accurate full-resolution behavioral videos, to our knowledge for the first time. We demonstrate the application of this framework to multiple behavioral paradigms and neural recording technologies.

A few exciting directions for future work are clear. First, for simplicity, in this work we decomposed our approach into individual compression, segmentation, and decoding steps. In principle it is possible to train the graphical model in Fig. 1 in an end-to-end fashion. This approach may lead to improved performance on compression and decoding metrics. Second, the ability to segment animal behavior into reproducible syllables opens up new possibilities for neural data analysis, for example, novel switching encoding models triggered by the segmentation output of the methods developed here [45]; these methods could also in principle be directly applied to the coordinates of tracked body parts from pose tracking algorithms. Finally, our unsupervised compression approach does not allow us to easily disentangle factors of variation in the behavior. For example, changes in arm position are generally represented across all latent factors, hindering our ability to connect neural activity with particular behaviors. Hybrid approaches that create a more interpretable representation of behavior, through the incorporation of labeled data from pose tracking algorithms, or from the timing of task-related variables (e.g., stimulus onset), seem particularly promising.

We hope to facilitate the application of these methods to a variety of behavioral datasets. A python implementation of our pipeline is available at https://github.com/ebatty/behavenet, which is based on the PyTorch [46], ssm [47], and Test Tube [48] libraries.

**Acknowledgments**   We thank N. Steinmetz, M. Carandini, and K. Harris for generously making their data publicly available. This work was supported by the Simons Foundation and the Gatsby Charitable Foundation, by NSF NeuroNex Award DBI-1707398, and by NIH awards 5U19NS107613, 5U19NS104649, and 1U19NS113201.

**Table 1:** Author contributions.

| | EB | MW | SS | DB | TA | SM | WG | JM | AC | JC | SD | SL | LP |
|---|---|---|---|---|---|---|---|---|---|---|---|---|---|
| Conceptualization | ■ | | | | | | | | | | ■ | ■ | ■ |
| Data collection | | | | | | ■ | | | | | | ■ | |
| Data analysis | ■ | ■ | ■ | ■ | ■ | | ■ | ■ | | | | ■ | |
| Code development | ■ | ■ | | | | | | | | | | ■ | ■ |
| Writing | ■ | ■ | | | | | | | | | | ■ | ■ |
| Editing | ■ | ■ | | | | | | | | ■ | | ■ | ■ |
| Funding acquisition | | | | | | | | | ■ | ■ | | ■ | ■ |

## Footnotes

†Joint senior authors

[1]Direct SVD was too slow due to the large matrices involved here, and randomized SVD approaches led to suboptimal results in our hands; instead, we simply used Adam to minimize the reconstruction MSE of a linear autoencoder (the same optimization problem solved by SVD/PCA), which uses a single dense layer for the encoder (images to latents) and a single dense layer for the decoder (latents to images), with encoding and decoding weights tied and a linear transfer function (with the same learning rate used for the nonlinear CAE).

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
