[Supplementary Material · behavenet_neurips19_final_appendix.pdf]

# Appendix A  Modeling details

## A.1  Data splits

We split data from each dataset into training, validation, and test trials. Training trials were used to fit model parameters; validation trials were used for early stopping in the CAEs and feedforward neural network decoders, as well as to choose the best model from hyperparameter grid searches (see Table A2). All figures and videos were produced using test trials.

The WFCI dataset combines two recording sessions from a single mouse (549 and 577 trials, each trial 189 frames at 30 Hz imaging rate). For each session we split blocks of 10 trials into 8 consecutive training trials, 1 validation trial, and 1 test trial. This 8-1-1 pattern was repeated for each dataset until all trials were assigned. For the NP and NP-zoom datasets we used a single recording session from a single mouse ("Waksman" in [18]). We arbitrarily defined "trials" as blocks of 1000 consecutive time points (~25 seconds), with a total of 96 trials. We split blocks of 10 trials into 5 consecutive training trials, 1 gap trial, 1 validation trial, 1 gap trial, 1 test trial, and 1 gap trial. The gap trials were conservatively inserted in order to reduce the similarity between neighboring training and validation/test trials, which we found could misleadingly increase the performance of the segmentation and decoding steps due to the lack of trial structure in this dataset (data not shown).

## A.2  CAE

To compress the behavioral videos we used a CAE architecture. An example architecture for the WFCI is shown in Table A1, which we hand designed. To test whether or not this architecture was a reasonable one we performed a limited CAE architecture search. We randomly generated 50 CAE architectures, allowing layer hyperparameters such as number of channels, kernel size, stride size, etc. to vary (the total number of layers was determined by these layer-wise hyperparameters). We fixed the learning rate at $1e$-4, the number of latents at 16, and we used no regularization. We then trained these 50 randomly generated architectures, as well as two hand designed architectures, for 20 epochs. We then fully trained the top 5 performing architectures using the early stopping criteria described in the main text. We found that the hand designed architecture shown in Table A1 was among the top 5 performing architectures (data not shown).

| Layer | Type | Channels | Kernel Size | Stride Size | Zero Padding | Output Size |
|---|---|---|---|---|---|---|
| 0 | conv | 32 | (5, 5) | (2, 2) | (1, 2, 1, 2) | (64, 64, 32) |
| 1 | conv | 64 | (5, 5) | (2, 2) | (1, 2, 1, 2) | (32, 32, 64) |
| 2 | conv | 256 | (5, 5) | (2, 2) | (1, 2, 1, 2) | (16, 16, 256) |
| 3 | conv | 512 | (5, 5) | (2, 2) | (1, 2, 1, 2) | (8, 8, 512) |
| 4 | dense | $N$ | NA | NA | NA | (1, 1, $N$) |
| 5 | dense | 64 | NA | NA | NA | (1, 1, 64) |
| 6 | reshape | NA | NA | NA | NA | (8, 8, 1) |
| 7 | conv transpose | 256 | (5, 5) | (2, 2) | (1, 2, 1, 2) | (16, 16, 256) |
| 8 | conv transpose | 64 | (5, 5) | (2, 2) | (1, 2, 1, 2) | (32, 32, 64) |
| 9 | conv transpose | 32 | (5, 5) | (2, 2) | (1, 2, 1, 2) | (64, 64, 32) |
| 10 | conv transpose | 2 | (5, 5) | (2, 2) | (1, 2, 1, 2) | (128, 128, 2) |

**Table A1:** Example CAE architecture for WFCI dataset using $N$ latents. Image sizes are (128, 128, 2) (one channel for each of side and bottom views). Kernel size and stride size are defined as (x pixels, y pixels); padding size is defined as (left, right, top, bottom); output size is defined as (x pixels, y pixels, channels).

## A.3  ARHMM

We fit ARHMMs with the EM algorithm, using the ssm package (https://github.com/slinderman/ssm). We initialized the discrete states using k-means, and then performed linear regression within each state to initialize model parameters. We found that although some variability in the final state sequence exists between different k-means initiaizations, these differences were usually quite small

(see errorbars in Fig. A3). For this work we only used a single autoregressive lag; using more lags may lead to better fits, but must be balanced with the amount of training data given the increase in number of model parameters (Fig. A3). We trained the models using 150 iterations of EM.

## A.4 Bayesian decoding

We develop a Bayesian approach to inferring discrete and continuous latent states. Typically, Bayesian decoding begins with a likelihood, $p(u \mid z, x)$, which models the conditional probability of neural activity $u$ given discrete states $z$ and continuous latents $x$. Combined with the prior $p(z, x)$ from the ARHMM, we would like to compute the posterior distribution $p(z, x \mid u)$. Unfortunately, defining a suitable likelihood model may be challenging when neural activity is high-dimensional, precedes or follows the observed behavior by an unknown delay, and has its own intrinsic dynamics.

Here we derive a Bayesian decoding method that circumvents the need for an explicit likelihood model, and instead relies on simpler neural network decoders that can be trained by standard approaches. The key idea is inspired by the "discriminative Kalman filter" of Burkhart et al. [44]. First, consider a simple hidden Markov model (HMM) with only discrete latent states. Assume that neural activity vectors $u_t$ and $u_{t'}$ are conditionally independent given the discrete states $z_t$, for any distinct time points $t \neq t'$. Then the posterior distribution on discrete states is

$$
\begin{aligned}
p(z \mid u) &\propto \prod_{t=1}^{T} p(z_t \mid z_{t-1})\, p(u_t \mid z_t), \\
&\propto \prod_{t=1}^{T} p(z_t \mid z_{t-1}) \frac{p(z_t \mid u_t)}{p(z_t)}\, p(u_t), \\
&\propto \prod_{t=1}^{T} p(z_t \mid z_{t-1}) \frac{p(z_t \mid u_t)}{p(z_t)}.
\end{aligned}
\tag{2}
$$

We used Bayes' rule to rewrite the likelihood in terms of the conditional probability $p(z_t \mid u_t)$, and since we are only interested in the conditional distribution of $z$, we dropped the constant $p(u_t)$. The marginal probability $p(z_t) = \sum_{z_{1:t-1}} p(z_{1:t})$ can be obtained by via the HMM filtering recursions, or approximated with the stationary distribution under the Markov transition kernel; we use the latter. The conditional probability $p(z_t \mid u_t)$ is approximated with a neural network, which is trained in a supervised manner using samples of discrete states and corresponding neural activity. With these approximations, we can compute posterior expectations of and draw samples from (2) using standard HMM message passing algorithms.

Of course, in the ARHMM, the neural activity is not conditionally independent given only the current discrete state. Integrating over the continuous states to obtain $p(z \mid u) = \int p(z, x \mid u)\, \mathrm{d}x$ would render the neural activity $u_t$ dependent on the entire history of discrete states $z_{1:t}$. History dependence and lags would further complicate the likelihood. Nevertheless, we find that the decoders described below, which use a window of neural activity $u_{t-\Delta:t+\Delta}$ to predict the discrete state $z_t$, are effective substitutes for the conditional distribution $p(z_t \mid u_t)$ necessary for the Bayesian decoder.

We follow the same technique to compute the conditional distribution $p(x \mid z, u)$ given neural activity $u$ and a discrete state path $z$. Here, the posterior is

$$
\begin{aligned}
p(x \mid z, u) &\propto \prod_{t=1}^{T} \mathcal{N}(x_t \mid A_{z_t} x_{t-1} + b_{z_t}, Q_{z_t})\, p(u_t \mid x_t, z_t), \\
&\propto \prod_{t=1}^{T} \mathcal{N}(x_t \mid A_{z_t} x_{t-1} + b_{z_t}, Q_{z_t}) \frac{p(x_t \mid u_t, z_t)}{p(x_t \mid z_t)}\, p(u_t \mid z_t), \\
&\propto \prod_{t=1}^{T} \mathcal{N}(x_t \mid A_{z_t} x_{t-1} + b_{z_t}, Q_{z_t}) \frac{p(x_t \mid u_t, z_t)}{p(x_t \mid z_t)}.
\end{aligned}
\tag{3}
$$

We approximate the conditional distribution $p(x_t \mid u_t, z_t)$ with a Gaussian distribution whose mean $\hat{x}_t$ and covariance $\hat{\Sigma}_t$ are obtained from the continuous state decoders described below. The covariance

of the Gaussian is set to the empirical covariance of the continuous state residuals. In theory, separate conditional distributions could be trained for each discrete state, but we find that a single distribution is sufficient. As with the discrete states above, we use a window of neural activity to predict $x_t$ instead of just a single frame $u_t$. Finally, we approximate the marginal probability $p(x_t \mid z_t)$ with a Gaussian distribution $\mathcal{N}(x_t \mid \tilde{\mu}_t, \tilde{\Sigma}_t)$ by using the Kalman filter to obtain the conditional means and covariances given the discrete state sequence.

For the derivation of the continuous posterior (3) to be valid, we need to enforce some constraints. Specifically, in order for the posterior to exist, $\hat{\Sigma}^{-1} - \tilde{\Sigma}_t^{-1}$ must be positive semi-definite. Therefore, we project this difference onto the positive semi-definite cone by performing an eigendecomposition and setting any negative eigenvalues to zero. We also found that multiplying $\tilde{\Sigma}_t$ by a positive constant, chosen by cross-validation, helped to stabilize decoder performance. Similarly, to obtain good results from the discrete state decoder we found that it was critical to have a sufficient amount of training data to estimate both the numerator and the denominator in (2); sample error in either of these quantities could lead to highly unstable results.

To summarize, we implement the Bayesian decoder by first sampling the discrete state sequence $z$ from (2) using discriminatively trained discrete conditional distributions. Then, we condition on the discrete state path and sample the continuous latent $x$ from the posterior (3), using the continuous Gaussian conditional distributions derived above. . The Bayesian decoder's continuous latent state estimate is obtained by $\hat{x}_{\text{Bayes}} \equiv \mathbb{E}[x \mid u] \approx 1/S \sum_{s=1}^{S} \mathbb{E}[x \mid z^{(s)}, u]$ where $z^{(s)} \sim p(z \mid u)$; we use $S = 100$. The inner expectation is computed via the Kalman smoother. All operations involve only linear-time message passing algorithms.

Next we describe the decoders for obtaining the conditional distributions $p(z_t \mid u_t)$ and $p(x_t \mid z_t, u_t)$.

### A.4.1 Decoding continuous states

To decode the CAE latents from neural activity we used a standard feedforward neural network $f_{\text{NN}}$, which minimized MSE between predicted ($\hat{x}_t$) and true ($x_t$) CAE latents using stochastic gradient descent. The input to the decoder was a window of neural activity centered at $t$ such that

$$\hat{x}_t = f_{\text{NN}}(u_{t-\Delta:t+\Delta}).$$

We performed a hyperparameter search on a session from each dataset, the details of which are shown in Table A2 (all intermediate layers used ReLU nonlinearities). The decoding architectures use the same number of units for each hidden layer, and 0 hidden layers corresponds to linear regression models. Finally, we set the decoder covariance to be the same for all time steps and equal to the covariance of the residuals $x_t - \hat{x}_t$. In theory, we could train a separate decoder for each discrete state $z_t$, but in practice we simply use one decoder regardless of the discrete state.

| | Hidden layers | Hidden unit number | Lags | $L2$ regularization |
|---|---|---|---|---|
| Search values | [0, 1, 3, 5] | [32, 64, 128] | [0, 2, 4, 8, 12, 16, 20, 24] | [1e-4, 1e-3, 1e-2] |

**Table A2:** Hyperparameter search details for CAE latent decoding and ARHMM state decoding.

### A.4.2 Decoding discrete states

We performed the same hyperparameter search when decoding ARHMM states from neural activity. The discrete state decoder minimized the cross entropy loss between predicted ($\hat{z}_t$) and true ($z_t$) ARHMM discrete states using stochastic gradient descent, where $z_t$ is the most likely state sequence.

## Appendix B    Supplementary figures

In order to verify that the discrete states inferred by the ARHMMs are meaningful we have shown that these states can be decoded from neural activity above chance (Fig. 3). Here we support the conclusion that the states are meaningful in several complementary ways: first by showing that the ARHMM generates reasonable samples of behavior (better than related HMMs and autoregressive (AR) models; see Fig. A1 and Supplementary Videos C.4), and second by showing that the inferred states are consistent across mice in the WFCI dataset (Fig. A2).

We also investigated how the amount of training data affects the model goodness-of-fit (as measured by the log-likelihood on held out data) as well as consistency of the model fit across multiple initializations (Fig. A3). To quantify the consistency of the model fits for a given amount of training data we computed the predicted CAE latents at each time point $t + 1$ as

$$\hat{x}_{t+1} = A_{z_t} x_t + b_{z_t}$$

where $z_t$ is determined from the most likely state path (i.e. the Viterbi path of the ARHMM). We then compare this 1-step ahead prediction between two models by measuring the MSE between them. This is a more robust measure than simply comparing the most likely state sequences for two models, especially for larger numbers of discrete states; for example, this 1-step ahead prediction MSE accounts for switches between states that might index very similar dynamics matrices. We calculate this metric by first training $N$ models on a given amount of training data, and then compute the 1-step ahead prediction MSE for all $N(N-1)/2$ pairs on held out data.

We show brain region-dependent results from decoding continuous CAE latents and discrete ARHMM states in the WFCI dataset, which demonstrates how these tools can be used to answer questions about how different representations of behavior are encoded across the brain. Decoder hyperparameters are the same as those in Tables A2 and A3.

Finally, we show decoding predictions on example trials from the NP and NP-zoom datasets. In Fig. 5 and the figures shown here, these trials were chosen based to be the most average trial (most similar to mean baseline, feedforward decoder, and Bayesian decoder MSE). The Bayesian decoder CAE latent predictions are fairly accurate for NP-zoom, and show reasonable uncertainty (Fig. A5). Interestingly, for the NP trial shown here, the discrete state decoder performs well, but the continuous state is not recovered accurately; nonetheless the pose of the mouse is reconstructed fairly well in the decoded image (Fig. A6). See Supplementary Videos C.3 for further examples and illustrations of the posterior decoding uncertainty expressed in image space.

**Figure A1:** Model-generated behavioral traces. For each dataset we generated behavioral latents from one of three models: an AR process, which can be thought of as an ARHMM with a single discrete state (*left column*); an HMM (*center column*), and the ARHMM (*right column*). For each dataset, the true CAE latents are shown on held out test data over time, with background colors indicating the discrete state inferred for that time step (*top row*); generated traces (conditioned on the above inferred discrete state) are also shown (*bottom row*). The AR model is able to capture temporal smoothness, but not diverse behavioral states (such as moving and still). The HMM model is able to capture more diverse behavioral states, but the lack of the AR process in the model structure leads to non-smooth states. The ARHMM is able to combine both of these elements to generate a diverse set of smooth latents. See Supplementary Videos C.4 for videos corresponding to each set of generated latents.

**Figure A2:** ARHMM state comparison across mice. Discrete states are shown for 24 trials of the WFCI dataset, using 9 CAE latents and 4 ARHMM states. Both mice exhibit a transition to a new (red) state after the levers move in (*red lines*), though mSM34 quickly transitions out of this state, perhaps indicating behavioral variability across mice. Additionally, both mice exhibit a transition to a new (blue) state after stimulus onset (*black lines*), and a transition to another (pink) state after reward spouts move in (*blue lines*). ARHMMs were trained separately for each mouse, and states were assigned colors based on location within the trial structure. Future work in this direction will fit a single ARHMM to multiple mice, thus allowing direct comparison of state usage across animals.

**Figure A3:** ARHMM model fits as a function of training data, CAE latents, and ARHMM states. We quantify the model goodness-of-fit using the ARHMM log-likelihood on held out test data (*top rows*), which tends to quickly saturate for smaller numbers of states while continuing to increase for larger numbers of states. We quantify model similarity across multiple initializations using the 1-step ahead prediction MSE between pairs of models (see text in Appendix B for details). The model fits tend to become more similar as the amount of training data increases. Together these metrics may be useful for determining how much data is necessary to achieve good, consistent model fits. Error bars reprsent standard error of the mean (SEM) across five initializations.

**Figure A4:** Non-Bayesian decoding performance as a function of brain region in WFCI dataset. We combined LocaNMF components [37] across hemispheres and areas into aggregate brain regions, e.g. components from all visual areas into a single 'VIS' region (*right column*). We then used these components to predict CAE latents (*left column*) and ARHMM states (*center column*) across two different sessions from a single mouse. We fit latents and states to behavior from both sessions simultaneously, and fit decoders to each session individually due to the session-dependent number of LocaNMF components.

**Figure A5:** Example decoded ARHMM states and CAE latents for a held out NP-zoom test trial. A) Discrete state probabilities inferred by the ARHMM for the behavioral data (*top row*); predictions from a feedforward model operating on neural data (*middle row*); output of the Bayesian decoding model (*bottom row*). B) CAE latents (*black*) are compared to predicted latents from a feedforward decoder (*blue*) and the Bayesian decoder (*green*). The shaded region indicates $\pm 3$ posterior SDs, output by the Bayesian posterior. C) Example frames of the real behavioral video (*top row*) compared to the Bayesian decoded frames (*bottom row*), see Supplementary Videos C.3 for full details.

**Figure A6:** Example decoded ARHMM states and CAE latents for a held out NP test trial. Conventions are as in Figure A5. Note that for this video the discrete state decoder A) is much more effective than the continuous latent decoder B), but nonetheless the decoded frames roughly capture the pose of the animal C).

# Appendix C    Supplementary videos

## C.1    Supplementary videos from Fig. 2 - CAE reconstructions

The videos show the original (downsampled) behavioral videos along with the reconstructions and the residuals from both the nonlinear CAE and the linear autoencoder using 8 latents. All these videos can be found at `https://drive.google.com/open?id=1QSXAzEkp3tDvLXYo11UIT7IPPnlFElRi`.

## C.2    Supplementary videos from Fig. 3 - ARHMM syllables

The videos show clips of the original behavioral videos corresponding to different examples of each discrete behavioral syllable (one syllable per panel). Clips begin with several frames that precede the transition to the given syllable, and the red square indicates the onset of the syllable. Clips are separated by several blank frames. For each dataset there is an example with K=2 syllables, which generally segment the behavior into still and moving. There is another example with K=16 syllables, which produce a more nuanced segmentation of the behavior. Syllables are ordered by their usage in the training data. All these videos can be found at `https://drive.google.com/open?id=1kn6fjyKRswNcjYU8zqJWCdPy2L9Be5Sk`.

## C.3    Supplementary videos from Figs. 4/5/A5/A6 - Bayesian decoding videos

The videos show decoding outputs from the fully Bayesian decoder. The center frame shows the true behavioral videos while the surrounding frames show samples from the posterior of the Bayesian decoder pushed through the CAE decoder into image space. The real CAE latents are displayed at the bottom, overlaid with each of the eight predicted latents. All these videos can be found at `https://drive.google.com/open?id=1QZiF9WUIvyxOjjQ3Ie6_6OCaMuMcLG5h`.

## C.4    Supplementary videos from Fig. A1 - ARHMM generated videos

The videos show clips of the CAE reconstructed behavioral videos next to clips of corresponding behavioral videos generated by the ARHMM. We sample the continuous latents conditioned on the inferred discrete states from the trained ARHMMs, then push these samples through the CAE decoder to produce conditionally synthetic behavioral videos. Along with samples generated by the ARHMM, we also show sample videos generated by an AR process (an ARHMM with K=1) and an HMM, which has temporal continuity in the discrete states but not the continuous latents. All these videos can be found at `https://drive.google.com/open?id=1Q-7FEOSBi8dIiBiQbyy9Vk1KSNLnZ883`.