[Reviews · NeurIPS 2019]

Reviewer 1



This paper proposes a probabilistic framework that combines non-linear (convolutional) autoencoders with ARHMM’s to model videos coming from neuroscience experiments. The authors then use these representations to build Bayesian decoders that can produce full-resolution frames based only on the neural recordings. I find the motivation of this paper - building tools to study the relationship between neural activity and behavior from a less reductionist approach - extremely valuable. I have, however, the following concerns: This work is very related to Wiltschko et al., the stronger difference being the use of nonlinear autoencoders instead of PCA. However, the difference between linear and non-linear AE in the reconstructions showed on the supplemental videos is not very noticeable. What are the units of MSE in Figure 2? How big is the improvement on decoding videos from neural data by using CAE as opposed to PCA in pixels? Secondly, in the paper the authors refer to “behavioral syllables” and while in Wiltschko et al there is an explicit mapping between states and behavioral subcomponents (walk, pause, low rear), this is missing in this paper. Can the resulting discrete states be mapped to any behaviorally interpretable state? In Fig. 3 the authors show that the sequence of states is task-related, can any of the states be related to task-related computations (left vs right responses, correct vs incorrect trials, etc)? Thirdly, in this paper the analyzed behaviors are very constrained - therefore is not very surprising that the videos can be encoded in 12-16 latent variables). Have the authors tried freely moving animals. Given that the whole idea is analyze behavior from a less constrained perspective, a less restricted behavior would be more interesting, in my view. [L210] “Nonetheless, many of the generated frames look qualitatively similar to real frames (Fig. 3D) and we can see the mouse transitioning between still and moving in a fairly natural way in the sampled videos, indicating that the generative model is placing significant probability mass near the space of real behavioral videos." This is, of course, really difficult to evaluate. But from the supplementary videos provided I’m concerned with: In NP many reconstructed video frames are static, without any movement, which doesn’t seem to be the case in the original video. Also, there are many segments with more than 2 paws reconstructed. In WFCI, the spouts are moving in the generative reconstructions, while they don’t move in the original frames. The decoding of videos purely from neural activity is a very interesting approach. One thing that confused me in supplementary video3_WFCI is the reconstruction of the two round structures on the lower part of the image. What is that and was that controlled by the animal? Small details: Reference to Fig. 1 ===UPDATE=== I had three main critiques in my review, which largely overlap with the points raised by the other reviewers: (1) Using a nonlinear auto-encoder doesn't seem to make a difference (at least qualitatively). (2) The inferred “behavioural syllables” are not interpretable. (3) The considered behaviours are too constrained. I have carefully read the author's feedback and appreciate the responses to all my points. However, I still believe that my points stand and therefore won’t change my score at this time: Point (1), the use of CAE, is one of the main novelties of the paper but is unclear how useful it is. The author’s claim that even if qualitatively we see no difference and the MSE is similar to that of a linear AE, it could help by reducing the number of parameters. However, as far as I understand, we have no evidence for that. More importantly, re point (2) the authors emphasise that the main point of the paper is “to provide a set of tools for understanding the relationship between neural activity and behavior”. However this is in contrast with their claim that ”interpretability of the behavioural syllables [...] is not our main concern; rather, we use the ARHMM as a prior model of behavior which is then incorporated into the Bayesian decoder”. As they later note, there are hints that these inferred states could be more closely related to the animal’s state (they have structure related to the task) but this is not followed in depth, in my view. I think this work is potentially very useful but could be greatly improved if the authors could show more explicitly how this method can provide more insights in the relationship between neural activity and behaviour.

Reviewer 2



The paper is very well written and the authors are honest about their results. The subject of the work is also very important to the community of neuroscience. Although the methods are not original, they are used very well together in the pipeline. The final results, however, do not look complete to me. The most important use of autonomous data analytics algorithms for science is to provide interpretable results, e.g. demixed data in Kobak et al. 2016, or interpretable components of TCA in Williams et al. 2018. To my understanding, presented framework does not provide any interpretable results. Even if not interpretable, the authors could show if the framework predicts the same state in a specific event in each trial, e.g. levers grab. I think figure 3c suggests that it does not. Also, regarding generating the video from neural data, the current results could be the product of learning the structure of the video and its changes over time, especially because very similar events happen in each trial (things that are different between trials such as animal's hands are not very well captured). In fact, the current results looks like average over all trials. I think it is important to see how using neural data actually improves the prediction (for example as opposed to average of all trials in each time step).

Reviewer 3



The paper clearly describes the method and is easy to read. The originality of the paper is in the composition of the CVAE+ARHMM modeling of animal behavior videos, combined with an SLDS decoding of behavior from neural activity. This is an impressive array of statistical methods, and I am excited to see how well it works. The biggest question is whether the model works well compared to simpler methods. It is nice to see qualitative results of the method in the supplemental videos.

[Author Response · NeurIPS 2019]

We would like to thank the reviewers for their thoughtful comments and questions.

The probabilistic framework we developed is intended to provide a set of tools for understanding the relationship between neural activity and behavior. These tools can compress, segment, and generate behavioral videos, as well as decode those videos from neural activity. We address reviewer concerns for these four tasks separately.

All three reviewers noted that compressing the videos with a convolutional autoencoder (CAE) did not seem to qualitatively outperform compression with a simple linear model (though the CAE did perform better quantitatively). We agree with this observation, and note that the use of the CAE is not critical to downstream analyses. However, by using the CAE we achieve the same MSE in pixel space as the linear model with fewer latents, thereby reducing the number of parameters in the subsequent segmentation and decoding models.

Reviewers 1 and 2 raised the concern that the discrete behavioral syllables inferred by the autoregressive hidden Markov model (ARHMM) were not interpretable, as has been demonstrated in previous work. First of all, we want to emphasize that interpretability of the behavioral syllables is only qualitative and is not our main concern; rather, we use the ARHMM as a prior model of behavior which is then incorporated into the Bayesian decoder. However, we agree that clear, nameable syllables are ultimately useful beyond decoding. We do in fact find some degree of interpretability in the behavioral syllables in the WFCI dataset, where a clear trial structure exists. In Fig. 3C we show the inferred behavioral syllables across many repeats of the trial. There is, for example, a syllable that almost always directly follows the lever grab (maroon), and another syllable that directly follows the spout movements (light blue). Reviewer 2 raised the concern that the syllable sequence is not the same across trials, but this could reflect trial-to-trial variability in animal behavior, which is typically not taken into account in standard analyses linking neural activity and behavior. Though we do not further pursue this variability in the manuscript (e.g. how behavioral variability is related to correct versus error trials) we think this is an extremely interesting application for these methods.

Reviewer 1 noted that the levers in the WFCI dataset (the round structures in the lower part of the video) are being reconstructed, which is indeed interesting. In this task, the mouse grabs the levers once they are moved closer. Our reconstructions indicate that information about the lever movement is present in the neural activity from which we are decoding, perhaps in a visual area.

Reviewers 2 and 3 suggested several comparison models to decode behavioral video from neural data. Reviewer 3 suggested fitting other models directly from neural activity to behavioral video, such as an RNN. If accepted, we will add this comparison to the manuscript (both quantitatively, through MSE in the pixel space, and qualitatively with reconstructed videos). Reviewer 2 also suggested comparing the decoding of the WFCI dataset to the trial-averaged video, given the stereotyped behavior introduced by the trial structure. We think some variant of this approach is a good comparison for understanding if variability in neural activity is actually able to decode variability in the behavior. Reviewer 3 asked how training this model end-to-end compares to the piece-wise training presented in our manuscript. We think this is an interesting question, and have already begun preliminary efforts in this direction, though cannot currently comment on the differences.

Finally, Reviewer 1 suggested the application of this approach to freely moving animals. We have in fact been doing this, and plan to share the results of that analysis in the future.

[Meta-Review · NeurIPS 2019]

I thank the authors for their submission. The paper presents a framework for analyzing behavioral videos coming by combining nonlinear autoencoders and ARHMM. I also thank the reviewers for their detailed and thoughtful comments and suggestions. The reviewers agree that the paper is well motivated and well written, however they also raise serious concerns about the quality and interpretability of the results. The reviewers make the following suggestions: 1. Please detail in the paper why using a nonlinear auto-encoder is important or beneficial, seeing as it qualitatively doesn't seem to make much of a difference in terms of performance. What are the benefits of using the CVAE? 2. The inferred “behavioural syllables” do not appear to be interpretable. The paper could be greatly improved if the authors could show more explicitly how the method can provide more insights in the relationship between neural activity and behaviour. At the very least, the authors can show that the behaviour syllables are stable / repeatably produced. 3. Please show / comment on how the system would behave in more complex settings. I strongly encourage the authors to take into account the reviewers' comments and concerns for the final manuscript, most importantly regarding the interpretability of the behavior syllables.